# Completing the Genome Sequence of *Chlamydia pecorum* Strains MC/MarsBar and DBDeUG: New Insights into This Enigmatic Koala (*Phascolarctos cinereus*) Pathogen

**DOI:** 10.3390/pathogens10121543

**Published:** 2021-11-25

**Authors:** Rhys T. White, Alistair R. Legione, Alyce Taylor-Brown, Cristina M. Fernandez, Damien P. Higgins, Peter Timms, Martina Jelocnik

**Affiliations:** 1Genecology Research Centre, University of the Sunshine Coast, Sippy Downs, Sunshine Coast, QLD 4557, Australia; rwhite1@usc.edu.au (R.T.W.); at21@sanger.ac.uk (A.T.-B.); ptimms@usc.edu.au (P.T.); 2Asia Pacific Centre for Animal Health, Melbourne Veterinary School, The University of Melbourne, Parkville, VIC 3010, Australia; legionea@unimelb.edu.au; 3Parasites and Microbes Programme, Wellcome Sanger Institute, Hinxton, Cambridgeshire CB10 1SA, UK; 4Sydney School of Veterinary Science, The University of Sydney, Sydney, NSW 2006, Australia; cfer0633@uni.sydney.edu.au (C.M.F.); damien.higgins@sydney.edu.au (D.P.H.)

**Keywords:** *Chlamydia pecorum*, comparative genomics, recombination, phylogenetics, strain DBDeUG_2018, strain MC/MarsBar_2018

## Abstract

*Chlamydia pecorum*, an obligate intracellular pathogen, causes significant morbidity and mortality in livestock and the koala (*Phascolarctos cinereus*). A variety of *C. pecorum* gene-centric molecular studies have revealed important observations about infection dynamics and genetic diversity in both koala and livestock hosts. In contrast to a variety of *C. pecorum* molecular studies, to date, only four complete and 16 draft genomes have been published. Of those, only five draft genomes are from koalas. Here, using whole-genome sequencing and a comparative genomics approach, we describe the first two complete *C. pecorum* genomes collected from diseased koalas. A de novo assembly of DBDeUG_2018 and MC/MarsBar_2018 resolved the chromosomes and chlamydial plasmids each as single, circular contigs. Robust phylogenomic analyses indicate biogeographical separation between strains from northern and southern koala populations, and between strains infecting koala and livestock hosts. Comparative genomics between koala strains identified new, unique, and shared loci that accumulate single-nucleotide polymorphisms and separate between northern and southern, and within northern koala strains. Furthermore, we predicted novel type III secretion system effectors. This investigation constitutes a comprehensive genome-wide comparison between *C. pecorum* from koalas and provides improvements to annotations of a *C. pecorum* reference genome. These findings lay the foundations for identifying and understanding host specificity and adaptation behind chlamydial infections affecting koalas.

## 1. Introduction

*Chlamydia pecorum*, a member of the family *Chlamydiaceae*, is a globally disseminated intracellular pathogen of livestock, free-range ruminants, and the koala (*Phascolarctos cinereus*) [1]. *C. pecorum* infections are ubiquitous in livestock, presenting with subclinical and/or asymptomatic shedding as well as a range of clinical diseases, including conjunctivitis, enteritis, mastitis, sporadic bovine encephalomyelitis (SBE), abortions and/or arthritis/polyarthritis [1]. In the iconic Australian tree-dwelling marsupial, the koala, *C. pecorum* infections have been known since the early 1970s to be the cause of keratoconjunctivitis and/or urinary tract infections which can lead to blindness or urinary incontinence and cystitis, respectively [2]. These diseases have been observed in koalas since the late 1800s and are continuing to pose a serious threat to their health [3,4].

To date, *C. pecorum* molecular studies utilising genotyping of the outer membrane protein A (*omp*A) and/or *C. pecorum*-specific multilocus sequence typing (MLST) have revealed important observations about infection dynamics and strain diversity within and between koala and livestock hosts [5,6,7,8,9,10,11,12]. These molecular analyses showed that the global *C. pecorum* population is diverse, but also that: (i) a single host can harbour two distinct genotypes at different anatomical sites [9,10]; (ii) genetically diverse genotypes can circulate in a single population [5,6,7,8,9,10]; (iii) the same genotypes can infect two different hosts (e.g., koala and sheep, sheep and goat, or sheep and cattle) [6,9,10,12]; (iv) some koala *C. pecorum* genotypes are more genetically similar to livestock *C. pecorum* strains than they are to other koala-derived *C. pecorum* [5,6,12]; and (v) genotypes infecting the northern Australian koala populations are genetically distinct from those infecting southern Australian koalas [5,8,11]. These traditional genotyping techniques are relatively inexpensive and prove valuable in surveillance and genetic diversity studies, but do not provide the resolution achieved using whole-genome sequencing (WGS). Meanwhile, genomic investigations are becoming a norm in bacterial studies, including for the genus *Chlamydia*, following recent innovations in WGS technologies which have greatly reduced costs [13,14,15,16]. The use of genomics combined with sample-specific metadata have provided important observations on the contemporary history of the human pathogen *C. trachomatis* as well as identifying current circulating lineages through populations and time [16].

Such approaches have been used for *C. pecorum* infections in livestock and koalas to understand the origin and genetic diversity, and to predict virulence factors of this enigmatic pathogen [17,18,19,20,21,22]. To date, these analyses have been performed using rather small both draft and/or complete genome collections in separate studies, leaving us with many unresolved questions particularly regarding the koala *C. pecorum* strains. In contrast to the variety of *C. pecorum* molecular studies, to date, only four complete and 16 draft genomes have been published. These include: three strains from European, Australian, and North American cattle, including the type strain E58 [18,19]; three strains from European pigs [20]; nine strains from European, North American, and Australian sheep [17,19,21,22]; and five strains from koalas [17,21]. Phylogenetic analyses of this limited koala *C. pecorum* strain genome collection revealed that *C. pecorum* genomes from koala populations in northern and southern Australia clustered into two distinct clades, although it must be noted that only one *C. pecorum* genome is available for the southern koala population [20,21]. Comparative genomics of strains from only three Australian northern koalas highlighted regions accumulating single-nucleotide polymorphisms (SNPs) and loci unique to koala-derived *C. pecorum* that may be associated with adaptation in these hosts [17]. Importantly, of the five available *C. pecorum* draft genomes collected from koalas, none are complete. Almost all are missing hallmark genomic regions associated with virulence in chlamydia infections [15,23], or have poor assemblies in these regions, such as the plasticity zone (PZ) and polymorphic membrane protein (*pmp*) genes, due to their repetitive nature. This further hinders our understanding of genome biology of the koala-derived *C. pecorum*. We should also note that the complete genome of the bovine E58 strain is used as the current reference genome for *C. pecorum* strains, including the distantly related koala *C. pecorum* strains. Using a distantly related genome as a reference may have additionally contributed to poor reference-assisted assemblies. Moreover, Gorrie and colleagues show that finer pair-wise SNP resolutions are achievable when using a closely related reference genome [24], which highlights the need for a complete genome of *C. pecorum* from koalas in future investigations. The complete reference koala-derived *C. pecorum* genomes could improve our current typing schemes, help us identify the regions with the most informative SNPs for accurately inferring lineages and/or even challenge our thinking about how we characterise *C. pecorum* from koalas.

A further challenge to obtain good quality chlamydial genomes is that members of the Chlamydiae class are strict obligate intracellular organisms that require mammalian or protozoan cells to grow, using laborious laboratory methods. Furthermore, clinical samples, in particular veterinary samples, often contain low bacterial loads, are collected in an absence of favourable growth media, and are often subjected to delayed transport to the laboratory, creating further challenges for chlamydial in vitro isolation [25,26]. To overcome these challenges, culture-independent molecular approaches using capture probes are attractive and widely used alternatives in the chlamydial fields [25,26]. However, the probe design also relies on complete reference genomes, and with increases in genomic diversity this can reduce the ability of the probe to capture regions of the target DNA [25,26].

Here, we report high-quality closed genomes of koala *C. pecorum* strains, DBDeUG_2018 (GenBank: CP080401 and CP080402) and MC/MarsBar_2018 (GenBank: CP080403 and CP080404) and re-evaluate their genomic make up. We expand on previous works by improving the gene annotations for DBDeUG and MC/MarsBar, and by providing the genomic context of important virulence and accessory gene content. Our dataset contains seven genomes (including these closed genomes) from koalas (*n* = 5 strains) and 13 genomes from livestock (*n* = 13 strains), with which we explore the evolutionary relationships between *C. pecorum* strains. After reconstructing the phylogeny, we confirm that recombination is common among *C. pecorum*. Moreover, we perform fine-detailed comparative genomics between koala *C. pecorum* strains to gain further insights into the virulence factors and diversity of koala-derived *C. pecorum*. This demonstrates the necessity for more genomic studies to resolve the emergence and dissemination of *C. pecorum* in koalas. Ultimately, these analyses provide reference genomes for future comparative genomics studies by describing the first closed genomes of *C. pecorum* associated with urogenital infections in the koala.

## 2. Results

### 2.1. High Quality Closed Genomes of C. pecorum DBDeUG_2018 and MC/MarsBar_2018

To fully resolve the genomes of *C. pecorum* DBDeUG and MC/MarsBar, isolates (renamed DBDeUG_2018 and MC/MarsBar_2018) were resequenced. WGS of DBDeUG_2018 and MC/MarsBar_2018 resulted in a total read count of 23,846,525, and 22,896,186, respectively. Read mapping to the genomes of E58, PV3056/3, P787, and W73 resulted with total mapped read counts of 16,458,765 (69.0% of total reads) and 13,336,552 (58.2% of total reads) for DBDeUG_2018 and MC/MarsBar_2018, respectively. Both genomes have a 100% genome coverage, while DBDeUG_2018 has a 3703.1-fold average sequencing depth whilst MC/MarsBar_2018 has a 2182.5-fold average sequencing depth. De novo assembly successfully resulted in a single circular chromosome of 1,106,377 bp and 1,106,403 bp for DBDeUG_2018 and MC/MarsBar_2018, respectively. As previously shown [27], both DBDeUG_2018 and MC/MarsBar_2018 carry a 7547 bp chlamydial plasmid (Table 1). Analysis of the plasmid to chromosome ratio analyses indicated that MC/MarsBar_2018 carries two plasmids per chromosome (ratio: 2 ± 0.24 standard deviation (SD)), whereas DBDeUG_2018 carries one plasmid per chromosome (ratio: 0.95 ± 0.06 SD). General features such as the number of coding DNA sequence (CDS) and GC content of these closed genomes are listed in Table 1, and are consistent with their draft counterparts, as well as other *C. pecorum* genomes. In silico, MLST confirmed both DBDeUG_2018 and MC/MarsBar_2018 as sequence type (ST)69.

The resequencing of *C. pecorum* MC/MarsBar_2018 and DBDeUG_2018 achieved 100% chromosome coverage, which can accurately denote the genomic content of these koala-derived reference strains (Figure 1, Table 1). Recent MC/MarsBar (GenBank: AZBC01000001 to AZBC01000014) and DBDeUG (GenBank: AZBB01000001 to AZBB01000008) draft genomes did not resolve complex genomic regions such as the PZ and *pmp* gene regions. When comparing the draft assemblies with our closed genomes, MC/MarsBar is missing at least 1250 bp and 11,000 bp in the major *pmp* gene cluster and the PZ, respectively (Figure 1a). DBDeUG is also missing sequences in the major *pmp* gene cluster (~4000 bp) and PZ (~8000 bp) (Figure 1b). Furthermore, using specialised nucleotide and amino acid sequence homology searches, we improved annotation for 104 of 301, and 109 of 305 previously annotated hypothetical genes for DBDeUG_2018 and for MC/MarsBar_2018, respectively. Notably, these analyses confirmed additional predicted metabolic genes, type III secretion system (T3SS) loci, inclusion membrane (*Inc*) genes and *pmp* genes.

### 2.2. Improved Annotations of C. pecorum Regions of Interest

With the complete genomes of MC/MarsBar_2018 and DBDeUG_2018, we were able to accurately denote *Inc*, *pmp*, and T3SS CDSs. We identified eight putative *Inc* genes, all with two transmembrane domains. Of those, MarsBar_0834 (putative Inc protein) and its homologue DBDeUG_0833 are located within the PZ and are *C. pecorum*-specific (Appendix A). Both closed genomes have 15 *pmp* CDSs, consisting of one each for genes *pmp*B, *pmp*D, *pmp*H, *pmp*E, outer membrane protein 11, two from *pmp*G/I with one of them being unique to *C. pecorum*, and eight of *pmp*G families (Appendix A). In total, we identified 26 T3SS genes in both genomes, consisting of 15 genes encoding putative T3SS apparatus components, seven genes encoding putative effectors and four for putative chaperones (Appendix A).

Next, we performed in silico predictive T3SS secreted proteins analyses using translated CDSs from both genomes using MC/MarsBar_2018 locus tags as reference. Of the seven predicted effectors, loci MarsBar_0284 (T3SS effector CDS), MarsBar_0303 (T3SS effector CT668 homologue CDS), MarsBar_0335 (homologue of the T3SS effector secreted inner nuclear membrane-associated *Chlamydia* protein (SINC)) and MarsBar_0445 (homologue of the T3SS effector translocated actin recruiting phosphoprotein (Tarp)) were predicted to be secreted according to the T3 secreted effector prediction analyses (Appendix A). Beside the four T3SS effectors described above, these analyses also predicted an additional 131 CDSs to be secreted (with high confidence scores of ≥0.9999). Of those, 53 CDSs were hypothetical proteins, 57 were metabolic genes CDSs, and the remaining 21 CDSs encoded *pmp*s, *Inc*s, known T3SS genes, outer membrane proteins, and lastly the plasmid virulence associated CDS5 or pGp3 (pCpecMarsBar_0005) (Appendix A). Notably, seven CDSs within the PZ were predicted to be secreted: MarsBar_0823, MarsBar_0827, and MarsBar_0836, encoding for hypothetical proteins; MarsBar_0829 and MarsBar_0833, both encoding for phospholipase D family proteins; and the intact MarsBar_0828, encoding for glycosyltransferase cytotoxin A (ToxA), as well as truncated MarsBar_0831, encoding for the glycosyltransferase sugar-binding region of glycosyltransferase cytotoxin B (ToxB) (Appendix A). It is interesting to note that the DBDeUG intact ToxA and ToxB (locus tags DBDeUG_0828 and DBDeUG_0831) are also predicted to be secreted (Appendix A). Finally, locus MarsBar_0613, encoding for the highly polymorphic tandem repeat ORF663 homologue was also predicted to be secreted.

Next, we have compared the previously denoted pseudogenes. Previously, premature stop codons were identified in two MC/MarsBar loci, namely, ToxB (CpecG_0814) and a hypothetical protein (CpecG_0412), and in DBDeUG hypothetical protein CpecF_0874. Our closed genomes also confirmed these genes are truncated (locus tags MarsBar_0831 and MarsBar_0430, the latter re-annotated as a Chromosome segregation ATPase, and locus tag DBDeUG_0889 (Appendix A)). In this study, we also observed additional premature STOP codon, for hypothetical protein CDS (locus tags MarsBar_0650 and DBDeUG_0651). The same is noted for MC/MarsBar and DBDeUG hypothetical protein CDS (loci CpecG_0638, and CpecF_0639, respectively), however these were not annotated as pseudogenes previously. Interestingly, the homologue hypothetical protein CDS in *C. pecorum* strain IPTaLE (locus tag CpecA_0640) also resulted in premature STOP codon, however, this was also not previously annotated. Compared to intact homologue in other *C. pecorum* genomes, CpecA_0640 was truncated at 39% of the gene length whereas loci MarsBar_0650 and DBDeUG_0651 were truncated at 84% of the gene length (Appendix A).

### 2.3. Phylogenetic Analyses of C. pecorum Confirm Separation of Livestock from Koala Strains

To determine the similarity between our resequenced koala *C. pecorum* genomes, the circularised complete chromosome sequences underwent a reference-free global alignment (1,106,861 bp). There were 2648 SNPs between MC/MarsBar_2018 and DBDeUG_2018. Of these 2648 SNPs, 36.6% (*n* = 970 SNPs) and 12.3% (*n* = 326 SNPs) are within the PZ (42,195 bp region spanning positions 897,543 to 939,738 in MC/MarsBar_2018) or major *pmp* gene cluster (36,886 bp region, spanning positions 590,974 to 627,860 positions in MC/MarsBar_2018), respectively. Outside of these regions, 1532 SNPs that separate MC/MarsBar_2018 from DBDeUG_2018 are distributed across the chromosome.

Next, the phylogenetic reconstruction of publicly available *C. pecorum* genomes allowed for the contextualisation of DBDeUG_2018 and MC/MarsBar_2018 amongst a global dataset (*n* = 20 genomes). Alignment of the core-genome of the 20 genomes (Appendix A) resulted in a 1,005,640 bp alignment, equating to 99.9% of the E58 chromosome. A total of 14,504 core-genome SNPs called against the reference chromosome E58 were identified and used to construct a maximum likelihood (ML) phylogenetic tree (Figure 2a). Based on tree topology and ≥95% bootstrapping, the phylogeny resolved into two well-supported and diverse clades. Clade 1 is comprised of strains collected from pigs (*n* = 3), cattle (*n* = 1), and a koala in South Australia (*n* = 1). In comparison, Clade 2 is comprised of strains collected from sheep (*n* = 7), koalas in South-East Queensland (SEQ) (*n* = 3) and New South Wales (NSW) (*n* = 1), and cattle (*n* = 2). Within Clade 2, strains collected from koalas (42% bootstrap) and livestock (100% bootstrap), clustered into their own sub-clades.

In Chlamydiae, the acquisition of genomic material from other species is considered uncommon [30]. Previous genomics studies have however, determined that recombination is common within *C. trachomatis* [31,32,33,34]. Previous *C. pecorum* WGS studies removed SNPs within recombination regions in a ~280 kbp core-genome alignment [20]. Thereby, to the best of our knowledge, regions of recombination have not yet been investigated in *C. pecorum*. To assess this and predict regions of recombination, we analysed the genome alignment using Gubbins. This identified seven major regions of recombination, with high SNP densities in and surrounding the major *pmp* gene clusters, T3SS genes, *omp*A region, and PZ (Figure 2a). Notably, a similar pattern of recombination was observed for the SEQ koala clade. Of the initial 14,504 core-genome SNP alignment, 10,479 (72.2%) SNPs were in regions of recombination and were removed from downstream analyses.

To determine tree topology, the remaining 4025 non-recombinant core-genome SNP alignment underwent phylogeny inference with ML and maximum parsimony (MP) to express branch lengths as the expected number of substitutions per site (Figure 2b) and the number of SNPs (Figure 2c), respectively. The MP and ML reconstructions using the 4025 non-recombinant SNP alignment resolved identical clustering. To validate the phylogenetic reconstruction, we included the draft genomes for DBDeUG and MC/MarsBar. Using the chromosome of E58 as a reference, DBDeUG and DBDeUG_2018 have no SNP differences in the core-genome alignment. Meanwhile, MC/MarsBar and MC/MarsBar_2018 differed by six SNPs (Appendix A). These six SNPs are however, in 16S ribosomal RNA (*n* = 2) and conserved chlamydial house-keeping genes (*n* = 4), which suggests a possible artefact or miss-assembly in the MC/MarsBar draft genome. Based on overall tree topology and bootstrap values ≥95%, the phylogeny can be split into four distinct clades (Figure 2c). The four genomes representing strains from koalas in SEQ (*n* = 3) and NSW (*n* = 1) cluster together in clade 2.4. Relative to E58, these four genomes were separated by a maximum pair-wise distance of 153 non-recombinant core-genome SNPs. These 153 SNPs are between DBDeUG_2018 (collected in Queensland in 2010) and Gun/koa1/Ure (collected in NSW in 2012) (Figure 2b,c). Further investigation reveals that these four genomes were distantly related with a median pair-wise distance of 77 (Interquartile range: 75 to 146) non-recombinant SNPs. Relative to E58, using the pair-wise SNP data, 40 and 37 SNPs define the terminal branches to DBDeUG_2018 and MC/MarsBar_2018, respectively (Appendix A).

### 2.4. Comparative Genomics with Other Koala-Derived C. pecorum Reveals New Loci of Interest Accumulating SNPs

To investigate genomic differences between koala-derived *C. pecorum*, the five koala *C. pecorum* genomes were analysed and compared. As evaluated by whole-genome alignments, the koala *C. pecorum* genomes are largely syntenic with similar predicted CDS counts (936 to 996) and order, and share >99.7% nucleotide identity (Appendix A). Most sequence variation was found in the same chromosomal regions as indicated above (such as T3SS genes, *omp*A, *pmp* clusters, and the PZ; Appendix A), although previous studies have shown the difficulty in accurately analysing these regions due to incomplete assemblies. In our resequenced genomes, where the PZ was well assembled, the PZs in MC/MarsBar_2018 and DBDeUG_2018 were conserved and syntenic with that of *C. pecorum* from livestock (Figure 3), with 96.5% nucleotide identity to E58, and 97.5% nucleotide identity to each other.

Next, we investigated nine polymorphic loci with 67.9 to 98.7% sequence identity between the strains from koalas, occurring as at least two haplotypes (Appendix A). For the loci MarsBar_0284, 0445, and 0842 we observed that 121, 233, and 54 SNPs respectively, separate the southern Australian strain SA/k2/UGT genes from highly similar homologues of the northern Australia koala strains (Appendix A). However, for the rest of the investigated loci, namely MarsBar_0283, 0320 (*omp*A), 0335 (T3SS SINC effector), 0564, 0593, and 0613 (ORF663 homologue) SNPs also appear to separate between northern Australian koala strains from NSW and SEQ (Appendix A). Most of these genes were under purifying and/or neutral selection with a d_n_/d_s_ ratio of ≤1 (Appendix A) with synonymous SNPs. Two loci, MarsBar_0842, encoding for inclusion membrane protein A, and MarsBar_0335, encoding for the T3SS SINC homologue, were identified to be under positive selection, having d_n_/d_s_ ratios of 1.77 and 1.95, respectively (Appendix A).

## 3. Discussion

### 3.1. Koala C. pecorum Genomics Revisited

*Chlamydia pecorum* remains an important global pathogen of livestock and koalas. Whilst we heavily rely on molecular studies to study the strain diversity and local epidemiology of this pathogen and the infections it causes, our understanding of the global population diversity, inter- and intra-host transmission, and virulence factors are in their infancy for this pathogen, especially compared to other related chlamydial species [11]. This investigation was motivated by the lack of high quality, closed koala *C. pecorum* reference genomes, and as such expands on previous descriptions from incomplete draft assemblies and resolves the genomic architecture of two *C. pecorum* ST69 strains collected from diseased koalas [15,23]. This new knowledge could improve our current typing schemes and even challenge the molecular characterisation of strains, particularly where two koala-derived *C. pecorum* strains with the same ST and/or *omp*A genotype are genetically different. Fine-detailed molecular characterisation of the strains is important for management of this organism in wild koala populations [6,7,8].

The *C. pecorum* genomics studies to date have mainly focused on detailed descriptions of genome content, namely, the number of *pmps*, T3SS and PZ genes, tryptophan and biotin operons, accumulation of SNPs and plasmid distribution, as well as phylogenetic relationships between livestock-derived and koala-derived *C. pecorum*, with more detailed analyses performed using complete livestock genomes [17,19]. The complete genomes of DBDeUG_2018 and MC/MarsBar_2018 were still largely congruent with these previous descriptions. However, in this study we provide new information on putative secreted effectors and denote genomic regions/loci of interest that could be useful for characterisation of *C. pecorum* strains obtained from koalas in an absence of WGS.

Previous *C. pecorum* genomic analyses identified one to six pseudogenes in the three koala and three livestock strains available [17,19], with these pseudogenes being unique to either the analysed koala or livestock *C. pecorum* and associated with adaptation to the cognate host. The low density of pseudogenes is common in *Chlamydia* spp., ranging between 9 to 12 in *C. trachomatis*, *C. muridarum*, and *C. suis* to 21 in *C. pneumoniae* and 28 to 29 in *C. abortus* and *C. psittaci*. In these species, truncations commonly occur in hypothetical proteins, but also in virulence-associated *pmps*, *Incs*, and cytotoxin genes [30]. In this study, in addition to the previously described truncations in two hypothetical proteins and *tox*B, we also observed an additional hypothetical protein pseudogene which is present in all three northern koala-derived genomes (locus tags MarsBar_0650, DBDeUG_0651 and CpecA_0640). In comparison, pseudogenes in livestock-derived *C. pecorum* genomes were mainly detected in the phospholipase D (PLD) genes in the PZ [19]. We expect that our observations will change with the inclusion of more koala and/or livestock *C. pecorum* genomes. The role of pseudogenes in *C. pecorum* remains unresolved and raises even more questions as to whether this is a result of an adaptation to new hosts (particularly relevant for koala-derived strains), tissue tropisms, or loss of virulence [15,23,30,36].

### 3.2. Our Effort to Broaden Understanding of C. pecorum Virulence Factors

Aside from the lack of comprehensive genomic studies, our understanding of *C. pecorum* virulence factors and thereby its pathogenic potential has also been hampered by the lack of cell biology functional studies. We observed in all *C. pecorum* genomes a functional Trp synthesis operon that contains (i) structural genes *trpABCDF* and *kynU*, and (ii) the repressor gene *trpR*. Islam et al. demonstrated in vitro that this intact operon provided *C. pecorum* with resistance to interferon gamma (IFN-γ), a key cytokine in chlamydial infection response [37]. Furthermore, in another study, Islam et al. also demonstrated that genetically distinct livestock *C. pecorum* isolates (including the genome-sequenced IPA, E58, and W73) display different in vitro growth phenotypes in different mammalian epithelial and immune cells [38]. This phenotypic variation may be associated with the identified variation in genomic content between *C. pecorum* livestock strains, however, this link certainly requires further investigation, particularly for koala-derived strains. In contrast to many genome biology studies of other related chlamydial species [39,40,41], comprehensive *C. pecorum* cell biology studies supplemented with genomics and/or “omics” are warranted.

To complete their developmental cycle and subvert host cell processes, *Chlamydia* spp. secretes a variety of proteins into the host cytoplasm which interact with the host cell cytosolic elements [39,41]. To broaden our understanding of the *C. pecorum* virulence factors, with a focus on T3SS effectors, this study utilised specialised nucleotide and amino acid sequence homology searches and predicted secreted effector analyses. In doing so, we identified 135 loci predicted to be secreted by *C. pecorum*. Of these, we identified four highly conserved CDSs encoding for T3SS effectors (loci MarsBar_0284 (T3SS effector CDS), MarsBar_0303 (T3SS effector CT668 homologue CDS), MarsBar_0335 (SINC homologue) and MarsBar_0445 (*Tarp* homologue) predicted to be secreted, according to the T3 secreted effector prediction analyses. While for two CDSs, locus tags MarsBar_0284 and MarsBar_0303, there were no matches to any of the homology or structure searches, the conserved *Chlamydia* T3SS effector SINC is considered a virulence factor due to its demonstrated ability to target the nuclear envelope of the infected as well as neighbouring cells [42]. In *C. pecorum*, intact SINC was identified in all genomes, encoding a 361-aa protein, with the gene being under positive selection (d_n_/d_s_ ratios = 1.95). This provides further evidence of a role of this protein in host tropism and pathogenesis. Similarly, it is well established that the chlamydial *Tarp* is a secreted T3SS effector that functions to remodel the host-actin cytoskeleton during the initial stage of infection [43]. In our study, ORF663 homologue (locus tag MarsBar_0613) was also predicted to be secreted, but there were no matches to any of the homology or structure searches. In previous molecular studies, *C. pecorum Tarp* and ORF663 genes or gene fragments have been denoted as valuable “neutral” and “positive selection contingency” markers, respectively, of *C. pecorum* intra-species phylogenetic analyses [44]. In our study, we also observed that highly polymorphic ORF663 and *Tarp* are under neutral selection.

Interestingly, of the remaining predicted secreted *C. pecorum* CDSs, the intact *tox*A, as well as both truncated (locus tag MarsBar_0831), and intact (locus tag DBDeUG_0831) *tox*B, were predicted to be secreted. To the best of our knowledge, this is the first predictive analysis for the *C. pecorum* cytotoxin genes. Previous studies indicated that the cytotoxin genes can be transcribed and produce functional proteins, and that *C. muridarum* cytotoxin mutant strains were less cytotoxic than the wild type [45]. The DXD-containing glycosyltransferases (which can transfer a range of different sugars to other sugars, phosphates, and proteins) could affect various host cellular processes, supporting its importance for the intracellular survival of chlamydia [46]. Comparisons with its homologue in *Escherichia coli*, adherence factor *efa1*, indicate that this potentially reduced cytotoxicity from the truncation of the *tox*B gene may lower the chlamydial burden during early infection [47]. This may relate to common presentation of subclinical and/or asymptomatic shedding, possibly contributing to the persistent nature of *C. pecorum*. In *C. pecorum*, the cytotoxin gene can be present in two copies in koala and livestock strains, but also up to three copies in porcine *C. pecorum* strains [20]. The exact role, activity of both truncated and intact cytotoxin genes, and their virulence potential requires extensive further investigation and should be a key focus for future functional studies.

### 3.3. Phylogenetic and Comparative Genomic Analyses of Koala C. pecorum Identify New Genetic Markers That Could Be Used for Intra-Species Fine-Detailed Molecular Epidemiology

While levels and regions of recombination across *C. pecorum* are similar to that observed in *C. trachomatis* [16] and *C. psittaci* [48], phylogenomic analyses indicate genetically distinct *C. pecorum* strains were found to be infecting northern (*n* = 4) and southern (*n* = 1) koala populations, compared to the *C. pecorum* from livestock hosts. Due to the fact we were limited to a small genomic collection, we observed lower levels of diversity than that reported using MLST [5,8,10]. Similarly, the origin of *C. pecorum* in koalas still remains open for debate. Previous MLST/*omp*A studies detected either the same *C. pecorum* genotypes in sheep and koalas [6,11,12] or koala genotypes more closely related to cattle *C. pecorum* than other koala genotypes [5,11], potentially indicating a contemporary spillover from livestock to koalas. An observation that *C. pecorum* in northern koalas form their own lineage (clade 2.4) hints to a potentially relevant evolutionary finding that the transfer of *C. pecorum* between livestock and koalas was perhaps a rare evolutionary event rather than a more contemporary event which has been hypothesised previously [11]. In contrast, *C. pecorum* from a koala in a southern population (SA/k2/UGT) clustered with strains from livestock in clade 1. This may be indicative of more contemporary *C. pecorum* spill-over from livestock to southern koala populations [5]. Regardless, more large-scale phylogenomic studies are needed to advance our knowledge about the emergence and dissemination of this enigmatic pathogen affecting koalas. Based on these observations, our methodologies propose a bioinformatics workflow for undertaking and reporting future *C. pecorum* genomic studies. With larger datasets, the inclusion of a closely related reference genome (the same ST and/or *omp*A genotype where possible) and the removal of sequences from regions of recombination will eventually allow for the description of possible transmission networks between koalas and other hosts of *C. pecorum*.

Using whole-genome trees, *C. pecorum* in northern koalas formed their own clade (clade 2.4), with some individual strains of the same STs branching separately. As WGS may not be available for some studies, we identified several candidate gene markers (such as the loci MarsBar_0283 (ChPn 76 kDa homologue), 0335 (SINC homologue), 0564 (conserved hypothetical protein CDS), 0593 (conserved hypothetical protein CDS), and 0613 (ORF663 homologue), in addition to *omp*A and selected *pmp* genes (such as Marsbar_0555 and Marsbar_0557) for future evaluation together with *C. pecorum*-MLST for *C. pecorum* intra-species fine-detailed molecular epidemiology. In *C. psittaci*, significant genetic variation in the T3SS effector and *pmp* loci also partially explained differences for host preferences and virulence characteristics for the *C. psittaci* genotypes [49].

### 3.4. Need for Further Research into the Role of Chlamydial Plasmids in C. pecorum Infections

Studies to date report that the *C. pecorum* plasmid (pCpec) is highly conserved and common in koala and livestock *C. pecorum* strains, with plasmid-free strains also occurring [27]. In our study, we estimated that *C. pecorum* strains from koalas seem to maintain the chlamydial plasmid at a low copy number according to plasmid to chromosome ratio analyses and predicted that the pCpec CDS5/pgp3 is secreted. While the role of the pCpec still remains widely open for investigation, previous studies denote this plasmid glycoprotein (CDS5/pgp3) as a major virulence factor involved in pathogenicity and infectivity [50,51]. A recent study describing WGS of *C. pecorum* strain 18-13680-18FL (ST23), isolated from one ovine abortion case revealed that the pCpec contained a unique deletion in CDS1/pgp3 that was also present in *C. pecorum* ST23 shed from the ewes, but not in other genetically distinct STs [52]. Though we did not find this deletion in our newly completed koala *C. pecorum* genomes, this new finding certainly raises further questions such as (i) whether this is also common in koala plasmid-bearing strains, or (ii) is it just a feature unique to livestock strains.

## 4. Materials and Methods

### 4.1. Chlamydia Pecorum Sample Description, Genome, Plasmid Copy Number Quantification, and Genome Sequencing

*C. pecorum* strain MC/MarsBar was isolated in 2009 from a female koala from Mount Cotton (27°37′ S, 153°13′ E) in SEQ, Australia, suffering from severe genital tract and ocular disease with chronic cystitis [44]. *C. pecorum* strain DBDeUG was isolated in 2010 from the urogenital tract of a wild female koala from Deception Bay (27°11′ S, 153°1′ E) in SEQ, suffering from a urogenital tract infection [17,44]. In previous studies, *C. pecorum*-specific MLST resolved MC/MarsBar and DBDeUG as ST69 [12], and *omp*A genotypes G and F, respectively [44], with WGS providing draft genomes for these isolates [17].

For this study, 500 µL aliquots of plaque-purified strains (previously passaged at least five times) were individually and routinely propagated in 90% confluent McCoy cells in T175 flasks in 2018, as described previously [53]. Following elementary body (EB) harvesting at 46 h post infection and centrifugation, 300 µL of purified EBs aliquots for each strain were heat-inactivated for 10 min at 90 °C. DNA extraction was carried out using a QIAamp DNA Mini Kit (Qiagen, Australia), according to the manufacturer’s instructions.

Prior to sequencing, DNA samples were quantified for chromosome and chlamydial plasmid copy number using the standard curve calibrated *C. pecorum*-specific qPCR assay targeting the 209 bp of the CpecG_0573 gene [54] and the 233 bp fragment of CDS5 of the pCpec [55], respectively. Briefly, all assays were carried out in a 15 µL total volume, consisting of 7.5 µL of iTaq master mix (Biorad, New South Wales, Australia), 0.5 µL of each 10 µM forward and reverse primer (Sigma Aldrich, Castle Hill, Australia), 4.5 µL of Milli-Q water (Merck Millipore), and 2 µL of DNA template. The qPCR assays were run for 35 cycles, and in each assay, positive (cultured plasmid-bearing *C. pecorum* MC/MarsBar DNA) and negative (mix only and Milli-Q water) controls were included. The *C. pecorum* chromosome and plasmid copy number in the two samples (tested in triplicate) were quantified by plotting the crossing points against a standard curve generated from triplicates of the ten-fold serial dilution of 106 to 100 copies/µL of previously quantified *C. pecorum* DNA and 233 bp CDS5 amplicon. Plasmid to chromosome ratio for each strain was estimated by dividing average plasmid copy number with average chromosome copy number.

DNA samples (renamed DBDeUG_2018 and MC/MarsBar_2018) were sequenced as paired-end 150 bp reads on the NextSeq 500 platform (Illumina Inc., San Diego, CA, USA) at the Australian Genome Research Facility (Parkville, Australia). DNA libraries were prepared following Illumina gDNA shotgun library preparation with bead size selection, following the manufacturer’s instructions.

### 4.2. Quality Control of Sequence Data

The FastQC package v0.11.9 (http://www.bioinformatics.babraham.ac.uk/projects/fastqc/, accessed on 3 November 2021) was used to assess quality metrics for the reads. To extract *C. pecorum* sequence data, paired-end reads were mapped to the chromosome of E58 (GenBank: CP002608) using the Burrows–Wheeler Aligner (BWA) v0.7.17 [56]. The mapped and unmapped reads were then converted from BAM format to FASTQ format using the bamtofastq function of Bedtools v2.30.0 [57]. Any unmapped reads were subsequently mapped to strains PV3056/3 (GenBank: CP004033), P787 (GenBank: CP004035), and W73 (GenBank: CP004034), and extracted. A similar approach was adopted for the plasmid sequence data, only using pCpecDBDeUG (GenBank: KT223770) and pCpecMarsbar (GenBank: KT223775) as reference genomes. All *C. pecorum* sequence data were compiled into a single FASTQ file for each of the forward and reverse reads. Sequence read data for DBDeUG_2018 and MC/MarsBar_2018 were submitted to the National Center for Biotechnology Information Sequence Read Archive under BioProject accession number PRJNA747116.

### 4.3. De Novo Genome Assembly, Genome Analyses, and In Silico Multilocus Sequence Typing

Prior to assembly, reads were conservatively trimmed and filtered using Trimmomatic v0.36 [58] to remove low-quality bases and Illumina adaptor sequences. Trimmomatic was used in paired-end mode with the following settings: a minimum quality of leading and trailing bases of 3, respectively; a sliding window trimming with a window size of 4 and a minimum average quality of 15; and minimum read length of 36 bp. The reads were de novo assembled using MGAP (https://github.com/dsarov/MGAP---Microbial-Genome-Assembler-Pipeline, accessed on 3 November 2021), which implements Velvet v1.2.10 [59]; VelvetOptimiser (https://github.com/tseemann/VelvetOptimiser, accessed on 3 November 2021); GapFiller v1.10 [60]; ABACAS v1.3.1 [61] (scaffolds against the chromosome of strain E58); IMAGE v2.4 [62]; SSPACE v2.0 [63]; Pilon v1.22 [64]; and MIRA v4 [65]. Each chromosome or plasmid was circularised by removing overlapping regions which were identified by performing self-comparative Nucleotide-Nucleotide BLAST (BLASTn) v2.9.0+ [66] searches (≥95% nucleotide identity) between the single contig before visualising using the Artemis Comparison Tool v18.1.0 [67]. The assemblies then underwent five rounds of additional polishing by mapping the corresponding Illumina reads to each contig using the BWA, and then correcting SNPs and insertions and deletions (INDELs) with Pilon.

Assemblies were annotated using Prokka v1.14.6 [68] using E58 as a reference for trusted proteins for the chromosomes. Annotations and the respective nucleotide sequences, particularly those denoting hypothetical protein genes, underwent further specialised searches using BLASTn, SmartBLAST [69], CD-Search [70], CDART [71], and were then manually corrected in Artemis v18.1.0 [72]. The MLST of DBDeUG_2018 and MC/MarsBar_2018 was determined in silico using MLST v2.19.0 (https://github.com/tseemann/mlst, accessed on 3 November 2021) with default settings to query the complete chromosomes against the Chlamydiales typing database hosted on PubMLST [28].

Further comparative genomic analyses were performed in Geneious Prime v2021.1.1 (http://www.geneious.com, accessed on 3 November 2021). The genomic regions of interest and/or polymorphic loci were extracted from the analysed genomes and/or aligned with progressiveMauve [73], the multiple alignments using fast Fourier transform (MAFFT) [74,75], and/or Clustal Omega (as implemented in Geneious Prime). The alignments were analysed for further nucleotide and/or translated protein sequence polymorphisms (such as the total number of polymorphisms (and gaps), percentage nucleotide sequence identity, number of haplotypes and haplotype diversity (Hd)) and ratios of the rates of non-synonymous to synonymous nucleotide substitutions per site (d_n_/d_s_) averaged over the gene alignment were calculated using DNASp 6.0 [76]. To predict the T3SS secreted proteins of *C. pecorum*, EffectiveDB (http://effectivedb.org, accessed on 3 November 2021) was used. For prediction, the standard Effective T3 classification module 2.0.1 was used with a high confidence cut-off score of 0.9999 [77]. Similarly, to predict transmembrane proteins, and identify *Inc* proteins characterised by bilobed hydrophobic domains using a cut-off of more than 40 amino acids in the bi-lobed transmembrane domain, TMHMM 2.0 server (https://services.healthtech.dtu.dk/service.php?TMHMM-2.0, accessed on 3 November 2021) was used.

### 4.4. Variant Detection and Phylogenetic Analyses

To estimate the number of SNPs and phylogenetic relationships of the two newly sequenced genomes, these were compared to a dataset representing *C. pecorum* genomes collected from cattle (*n* = 3), pigs (*n* = 3); sheep (*n* = 7), and koalas (*n* = 5) (Appendix A). Parsnp v1.2 [78] was used to generate a core-genome alignment (alignment of the syntenic regions across all genomes). The SNP density plots were visualised using the ggplot2 v3.3.5 library [79] in the R package v4.1.0 [80]. The Gubbins algorithm v2.3.4 [81] (default settings, “raxml mode” with the General Time Reversible (GTR) GAMMA correction) was used to identify regions of recombination. Resulting SNP alignments were used to reconstruct phylogenies. MP trees were reconstructed using the heuristic search feature of PAUP v4.0a [82]. ML phylogenetic trees were reconstructed using RAxML v8.2.12 [83] (GTR-GAMMA correction) thorough optimisation of 20 distinct, randomised MP trees, before adding 1000 bootstrap replicates. The resulting phylogenetic trees were visualised using FigTree v1.4.4 (http://tree.bio.ed.ac.uk/software/figtree/, accessed on 3 November 2021). The recombination blocks, aligned with the assembly-based ML phylogeny generated from the alignment of core-genome SNPs, were visualised in Phandango v1.3.0 [84].

## 5. Conclusions

In conclusion, we present the first two complete, closed, *C. pecorum* genomes sequenced from diseased koalas. These genomes should be used as high-quality reference sequences for future comparative genomics studies of *C. pecorum*. Furthermore, using these genomes, we expand on previous works by improving the gene annotations and by providing the genomic context of important virulence and accessory gene content. We confirm that recombination is common among *C. pecorum* and recommend identifying regions of recombination in future investigations to avoid inaccurate vertical phylogenetic inferences caused by frequent lateral gene transfer. These findings lay the foundations for identifying and understanding species specificity and host adaptation behind chlamydial infections affecting koalas. More large-scale WGS experiments and detailed epidemiological data is needed to ultimately enable health interventions.

## Figures and Tables

**Figure 1 pathogens-10-01543-f001:**
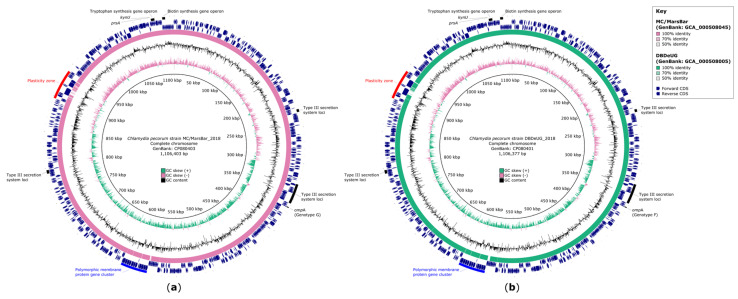
Genomic comparisons between *Chlamydia pecorum* MC/MarsBar_2018 and DBDeUG_2018, and their respective draft assemblies. (**a**) Circular representation of the chromosome of *C. pecorum* strain MC/MarsBar_2018 (GenBank: CP080403). (**b**) Circular representation of the chromosome of *C. pecorum* strain DBDeUG_2018 (GenBank: CP080401). The innermost rings represent the position in the genome in base pairs, GC skew, and GC content. Ring 4 represents nucleotide identity between sequences according to BLASTn (50 to 100%) between the complete chromosome and respective draft assembly. The coding sequence (CDS) for the forward (ring 5) and reverse (ring 6) strand is plotted in navy. Ring 7 highlights regions of interest. Circular genome plots created using BRIG v0.95 [29].

**Figure 2 pathogens-10-01543-f002:**
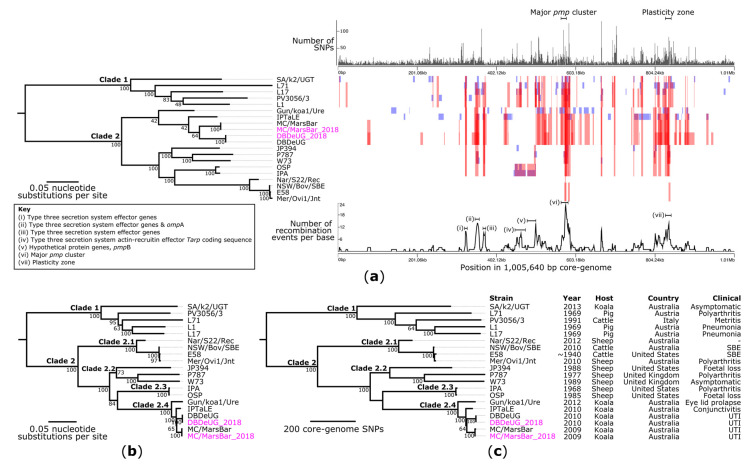
Evolutionary reconstruction of *Chlamydia pecorum*. (**a**) Maximum likelihood phylogeny inferred from 14,504 core-genome single-nucleotide polymorphisms (SNPs) from 20 genomes prior to recombination filtering. SNPs were derived from a core-genome alignment of 1,005,640 bp and are called against the reference chromosome E58 (GenBank: CP002608). The core-genome phylogeny (left) is plotted against a presence/absence matrix of regions of recombination predicted by Gubbins (right). Blue blocks represent recombination unique to that isolate, whereas red blocks represent ancestral recombination, shared by multiple strains. The graphs above and below the matrix represent the number of SNPs in a 1000 bp window and number of recombination events per base, respectively. (**b**) Maximum likelihood and (**c**) maximum parsimony phylogenies inferred from 4025 non-recombinant core-genome SNPs, from 20 strains, called against E58. Phylogenies are midpoint rooted which corresponds to the actual root by *C. pneumoniae* strain AR39 (GenBank: AE002161), which has been omitted for visualisation. Branch lengths represent the nucleotide substitutions per site (**a**,**b**) the number or core-genome SNPs (**c**), as indicated by the scale bar. Bootstrap values (using 1000 replicates) are shown. The consistency index for the tree shown in panel (**c**) is 0.8.

**Figure 3 pathogens-10-01543-f003:**
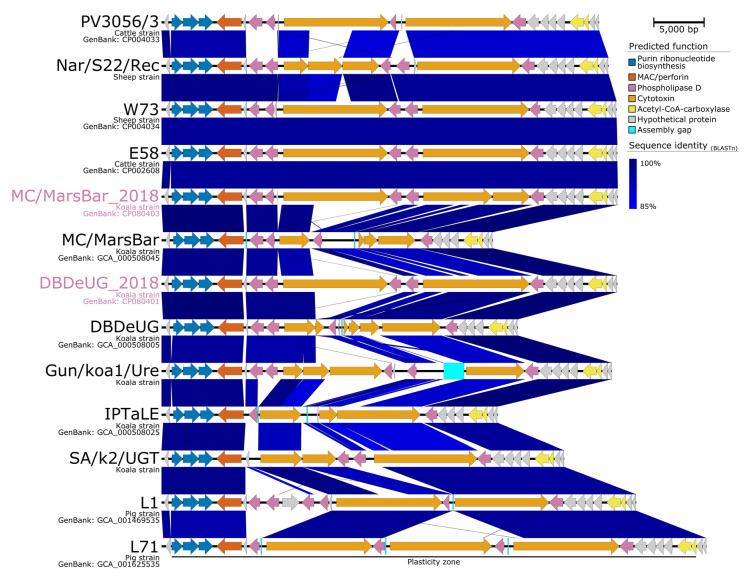
Major structural features and nucleotide pair-wise comparisons of the plasticity zones in *Chlamydia pecorum*. Blue shading indicates nucleotide identity between sequences according to BLASTn (85 to 100%). Key genomic regions are indicated. Strains Nar/S22/Rec, NSW/Bov/SBE, Gun/koa1/Ure, IPTaLE, SA/k2/UGT, L1, and L17 are represented as draft genomes. Image created using Easyfig v2.2.2 [35].

**Table 1 pathogens-10-01543-t001:** Genomic descriptions of resequenced *Chlamydia pecorum* DBDeUG_2018 and MC/MarsBar_2018 representative genomes from the koala.

Characteristics	*Chlamydia pecorum* Strain				
	E58	DBDeUG	DBDeUG_2018	MC/MarsBar	MC/MarsBar_2018
Multilocus sequence typing *^a^*	ST23	ST69		ST69	
*omp*A genotype	E58 *omp*A	F		G	
Collection year	Circa 1940	2010		2009	
Location	United States	Australia		Australia	
Host	Cattle	Koala		Koala	
Clinical presentation	Sporadic bovine encephalomyelitis	Urinary tract infection		Chronic cystitis	
Anatomical site	Brain	Urogenital tract		Urogenital tract	
BioSample	SAMN02603383	SAMN02470821	SAMN20256222	SAMN02470823	SAMN20256223
GenBank accession numbers (chromosome and plasmid)	CP002608	AZBB01000001 to AZBB01000008	CP080401 & CP080402	AZBC01000001 to AZBC01000014	CP080403 & CP080404
SRA accession	Not publicly available	Not publicly available	SRR15170908	Not publicly available	SRR15170907
Total No. of reads	Not applicable	13,889,239	23,846,525	15,011,176	22,896,186
No. of mapped reads	Not applicable	-	16,458,765	-	13,336,552
Chromosome length	1,106,197 bp	1,092,392 bp	1,106,377 bp	1,090,698 bp	1,106,403 bp
Predicted CDS	988	938	939	940	936
GC content (%)	41.11	41.12	41.12	41.12	41.11
No. of tRNA genes	38	38	38	38	38
No. of rRNA operons	3	3	3	3	3
No. of pseudogenes	1	1	2	2	3
Plasmid length	Absent	7547 bp	7547 bp	7547 bp	7547 bp
Predicted CDS	Not applicable	8	8	8	8
GC content (%)	Not applicable	31.55	31.55	31.55	31.55
Tandem repeats	Not applicable	4 × 22 bp	4 × 22 bp	4 × 22 bp	4 × 22 bp

*^a^* Against the Chlamydiales typing database hosted on PubMLST [28].

## Data Availability

Genome sequence data for the Australian strains in this study are deposited on the National Center for Biotechnology Information (NCBI) under BioProject accession number PRJNA747116. *Chlamydia pecorum* Illumina sequence read data are deposited on the sequence read archive (SRA) under the accession numbers SRR15170907 and SRR15170908. The complete genomes of DBDeUG_2018 and MC/MarsBar_2018 have been deposited to GenBank (accession numbers: CP080401 to CP080402 and CP080403 to CP080404, respectively). The programs used to analyse raw sequence reads for polymorphism discovery and whole-genome sequencing based phylogenetic reconstruction are available as described in the materials and methods. The authors confirm all supporting data, code, and protocols have been provided within the article or through Appendix A.

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
