# Peer review of "Completing the Genome Sequence of Chlamydia pecorum Strains MC/MarsBar and DBDeUG: New Insights into This Enigmatic Koala (Phascolarctos cinereus) Pathogen"

_pathogens, 2021, doi:10.3390/pathogens10121543_

Round 1

Reviewer 1 Report

Solid work, provides significant advancement to the field.

Inconsistent use of italics for bacterial names, please fix

Author Response

We thank the reviewer for this positive comment about our paper.

We have fixed the bacterial names throughout the manuscript. 

Reviewer 2 Report

In this study Authors describe the first two complete C. pecorum genomes from diseased  koalas. The manuscript is  exhaustive, highly topical and its content is worth being published.

Minor remarks:

Results: some sentences would be better placed in the discussion (line 236-241, 285-286)

Discussion

Line 422-425: this result could deserve a more detailed epidemiological comment

Please standardize the wording in italics in the text (e.g. C. pecorum, pmp...)

Author Response

We thank the reviewer for this positive comment about our paper, and we have addressed all comments as suggested.

Minor remarks:

Results: some sentences would be better placed in the discussion (line 236-241, 285-286)

Authors response: We do agree with the reviewer, however in order to give more context and clarity to this analysis and results we wish to keep them in this section.

Discussion

Line 422-425: this result could deserve a more detailed epidemiological comment

Authors response: Thank you for this insightful comment. To better place in context our discussion, we have added following comment:

Lines 419 – 423: “While levels and regions of recombination across C. pecorum are similar to that observed in C. trachomatis [16] and C. psittaci [48], phylogenomic analyses indicate genetically distinct C. pecorum strains found infecting northern (n = 4) and southern (n = 1) koala populations, and compared to C. pecorum from livestock hosts. Similarly, origin of C. pecorum in koalas still remains open for debate. Previous MLST/ompA studies detected either the same C. pecorum genotypes in sheep and koalas [6, 11,12] or koala genotypes more closely related to cattle C. pecorum than other koala geno-types [5,11], indicating on contemporary spillover from livestock to koalas.”

Please standardize the wording in italics in the text (e.g. C. pecorum, pmp...)

Authors response: Thank you for this comment. Somehow this was omitted, and has been corrected thorough the manuscript.

Reviewer 3 Report

In this manuscript, White et al. describe complete genomes of two Chlamydia pecorum strains from koalas. This study basically supplements previous studies that had only reported incomplete genome of the strains used in this study. Overall, this study seems to provide complete genomes koala infecting C. pecorum strains that could be used as reference genomes for future studies and also identifies gene loci that could be used for molecular epidemiology.

Major comment:

  1. The authors used strains that were collected several years ago and were passaged in lab (as mentioned in methods). How many time these strains were passaged in cell cultures? and could their passage in tissue culture (different than being in actual host) have an affect on genome?
  2. The sample size for phylogenetics analysis is low. The authors should be cautious in interpreting these results, especially geographical separation of the strains. 

Minor comments:

  1. Please check for bacterial name. C. pecorum has been used in full at some places where the genus name should have been abbreviated. Also, throughout the results section the bacterial name has not been italicized.
  2. Also look for gene names uniformity (ORF663 has been named differently in different places).
  3. Check numbering of results section.
  4. Finally, check the sequence of Supplementary figures. The figure S3 appears before S2 in the text.

Author Response

We thank the reviewer for the positive and very constructive comments about our paper. We have addressed all comments as suggested and to the best of our ability.

Major comment:

The authors used strains that were collected several years ago and were passaged in lab (as mentioned in methods). How many time these strains were passaged in cell cultures? and could their passage in tissue culture (different than being in actual host) have an affect on genome?

Authors response: To the best of our knowledge, these strains have been passaged at least 5 – 7 times since their isolation (as a part of only few cell biology/vaccine studies). In each passage, we generate working stocks that used for downstream analyses. We have used ~1 X 10*6/mL plaque-purified EBs (from 5th – 7th passaged master stock) and 500ul of these have undergone one replicative cycle in McCoys grown in T175 Flasks (to achieve high yield for WGS, equivalent to 1x10*6 genome copies/ul according to qPCR), followed by DNA extraction and WGS.

Previous studies have shown that chlamydial genomes remain constant and stable under multiple in vitro passaging over several years, as well as in vivo. Compared to previous draft DbDEUg, and Marsbar genomes, our genomes had none for DbDEUg and only 6 SNPs difference for Marsbar (believed to be an artefact of previous assembly and WGS technology).

Please see: https://www.ncbi.nlm.nih.gov/pmc/articles/PMC3417540/

https://www.frontiersin.org/articles/10.3389/fcimb.2012.00068/full

https://academic.oup.com/jid/article/215/11/1657/3079217

We have added this in text, see lines 471 – 476:

 “For this study, 500 µl aliquots of plaque-purified strains (previously passaged at least five times) were individually and routinely propagated in 90% confluent McCoy cells in T175 flasks in 2018, as described previously [53]. Following elementary bodies (EBs) harvesting at 46 h post infection and centrifugation, 300 µl of purified EBs aliquots for each strain were heat-inactivated for 10 minutes at 90C. DNA extraction was carried out using a QIAamp® DNA Mini Kit (Qiagen, Australia), according to the manufacturer’s instructions.”

Thank you for this great comment. However, it is very possible that these strains would accrue (we would hypothesise very minor) polymorphisms when taken from hosts, and then re-isolated in cell culture. Indeed, sequencing directly from the clinical swab/tissue and then comparing it to the sequencing of the in vitro isolate would resolve this, and it would be great experiment. Investigations like this are warranted in the future.

The sample size for phylogenetics analysis is low. The authors should be cautious in interpreting these results, especially geographical separation of the strains.

Authors response: Thank you for this comment. We do agree, especially as we have only 1 koala strain from South Australia, and 4 from Northern Australia (Qld and NSW). This highlights the immediate need for WGS of many geographically separated koala stains to see whether this observation still stands.

According to your comment, we have revised and removed biogeographical separation wording:

Line 213, section 2.3 subtitle:” Phylogenetic analyses of C. pecorum confirm separation of livestock from koala strains”

Lines 414-418, section 3.3 discussion: While levels and regions of recombination across C. pecorum are similar to that observed in C. trachomatis [16] and C. psittaci [48], phylogenomic analyses indicate on genetically distinct C. pecorum strains found infecting northern (n = 4) and southern (n = 1) koala populations, and C. pecorum from livestock hosts. Due to the fact we were limited to a small genomic collection, we observed lower levels of diversity than that reported using MLST [5, 8, 10]. Similarly, the origin of C. pecorum in koalas still remains open for debate. Previous MLST/ompA studies detected either the same C. pecorum genotypes in sheep and koalas [6, 11,12] or koala genotypes more closely related to cattle C. pecorum than other koala genotypes [5,11], potentially indi-cating on contemporary spillover from livestock to koalas.

Minor comments:

Please check for bacterial name. C. pecorum has been used in full at some places where the genus name should have been abbreviated. Also, throughout the results section the bacterial name has not been italicized.

Authors response: Thank you for this comment. Somehow this was omitted, and has been corrected thorough the manuscript.

Also look for gene names uniformity (ORF663 has been named differently in different places).

Authors response: This has now been corrected thorough the manuscript.

Check numbering of results section.

Authors response: Thank you for spotting this. The result sections are now corrected and numbered in sequence.

Finally, check the sequence of Supplementary figures. The figure S3 appears before S2 in the text.

Authors response: This has now been corrected thorough the manuscript.